# Adding links on minimum degree and longest distance strategies for improving network robustness and efficiency

**Masaki Chujyo** [ID]<sup></sup>*, **Yukio Hayashi**

Graduate School of Advanced Science and Technology, Japan Advanced Institute of Science and Technology, Nomi, Ishikawa, Japan

☉ These authors contributed equally to this work.
* mchujyo@jaist.ac.jp

## Abstract

Many real-world networks characterized by power-law degree distributions are extremely vulnerable against malicious attacks. Therefore, it is important to obtain effective methods for strengthening the robustness of the existing networks. Previous studies have been discussed some link addition methods for improving the robustness. In particular, two effective strategies for selecting nodes to add links have been proposed: the minimum degree and longest distance strategies. However, it is unclear whether the effects of these strategies on the robustness are independent or not. In this paper, we investigate the contributions of these strategies to improving the robustness by adding links in distinguishing the effects of degrees and distances as much as possible. Through numerical simulation, we find that the robustness is effectively improved by adding links on the minimum degree strategy for both synthetic trees and real networks. As an exception, only when the number of added links is small, the longest distance strategy is the best. Conversely, the robustness is only slightly improved by adding links on the shortest distance strategy in many cases, even combined with the minimum degree strategy. Therefore, enhancing global loops is essential for improving the robustness rather than local loops.

## Introduction

Our social, productive, and economical activities are supported by many complex networks, such as the Internet, World Wide Web, electric power, transportation, water supply, and international trading systems. These real-world networks have commonly scale-free (SF) structures characterized by power-law degree distributions [1], and therefore are extremely vulnerable against malicious attacks [2]. On the other hand, the robustness of connectivity is strongly related to both the degree distributions and the types of node removals, e.g. malicious attacks or random failures. By the attacks, nodes are removed in the decreasing order of degrees or other topological properties such as betweenness or closeness centralities, while by random failures, nodes are removed uniformly at random. Unfortunately, many SF networks are vulnerable against malicious attacks but tolerant against random failures in comparison with

**Data Availability Statement:** AirTraffic is available from http://konect.cc/networks/maayan-faa/. Email is available from https://deim.urv.cat/~alexandre.arenas/data/welcome.htm. Yeast is available from

http://vlado.fmf.uni-lj.si/pub/networks/data/bio/Yeast/Yeast.htm.

**Funding:** YH is partially supported by a Grant-in-Aid for Scientific Research (Grant Number JP.21H03425) from the Japan Society for the Promotion of Science. The funders had no role in study design, data collection and analysis, decision to publish, or preparation of the manuscript. There was no additional external funding received for this study.

**Competing interests:** The authors have declared that no competing interests exist.

synthetic Erdős-Rényi random graphs [2]. To overcome the vulnerabilities in real-world networks, we consider how to reconstruct more robust networks against malicious attacks, because a simple recovery only restores the original fragile structure.

In the state-of-the-art, an onion-like structure is known as the optimal robust network against attacks [3–5]. The onion-like structure has a positive degree-degree correlation [6], in which nodes with similar degrees tend to be connected. It is constructed by random rewiring in maximizing a robustness index against high degree adaptive attacks [3]. Moreover, several rewiring methods have been proposed for improving the robustness by selecting rewired nodes to increase the degree-degree correlations [7, 8].

Recently, apart from the degree-degree correlation, we remark that loops are deeply related to robustness against attacks [5, 9, 10]. In particular, the asymptotic equivalence of network dismantling and decycling has been derived [9]. Here, network dismantling problem is to find a minimum fraction of nodes whose removal makes the size of connected components into a given size, while network decycling or Feedback Vertex Set (FVS) problem is to find a minimum fraction of nodes whose removal makes the network no loops. Intuitively, a network without loops is fragile as a tree, because it is easily fragmented by any node removals. Thus, as a network is hard to become a tree by node removals, it is more robust. Furthermore, the relation between enhancing loops and robustness is also discussed for generating onion-like networks [5]. In the incrementally growing methods, a node is added at each time step in connecting to a randomly selected node and to a node with the minimum degree within a few hops from the randomly selected one. Since the attachments make new interwoven loops or bypasses, the incrementally growing method generates an onion-like network with both high robustness and a large size of FVS. Thus, it has been shown that the robustness and the size of FVS are strongly correlated [5]. Moreover, for improving the robustness, an effective rewiring method has been proposed by selecting nodes to increase the size of FVS instead of the degree-degree correlation [11]. However, the effects of the length of loops on the robustness are rarely discussed. Several recent studies suggest that long-range loops contribute to being robust more than short-range loops [12–15]. By a percolation analysis, it has been found that the robustness becomes weak due to short loops with a constant length in geographic networks [12, 13].

This paper considers link addition methods for improving the robustness of real networks in distinguishing the effects of long and short loops. Instead of rewiring which removes existing links, link additions can improve robustness without destructing the existing structures. Previous works [14, 16–20] have been discussed some link addition methods to improve the robustness. The basic methods repeatedly add links to pairs of unconnected nodes chosen uniformly at random. This random additions improve the robustness of SF [16] and geographical networks [17]. However, over the last decade, two effective strategies have been further proposed. Based on the minimum degree [16, 18–20] or the longest distance strategies [14], they select pairs of unconnected nodes to add links. It has been numerically shown that the minimum degree strategy improves the robustness more than the basic random additions in the same number of adding links [16]. On the other hand, it has been numerically shown that the longest distance strategy outperforms other methods on the robustness [14].

In networks with more complicated structures, link addition methods have also been discussed [21–23]. For improving the robustness of both interdependent and multiplex networks, the minimum degree strategy is effective [21]. Furthermore, the minimum degree strategy is also effective for enhancing the robustness against cascade failures induced by both random failures and targeted attacks in interdependent power grids and communication networks [22]. The minimum degree strategy is considered an effective method for interdependent and multiplex networks, while the longest distance strategy is effective for spatial networks. In spatial networks, adding many links or long links can improve robustness against attacks [23].

Although there are several discussions about effective link addition methods in various types of networks, they are basically based on either the minimum degree or longest distance strategies.

Thus, we investigate the contributions of the minimum degree and longest distance strategies to improving robustness against attacks by adding links, respectively. We point out that the effects of the minimum degree and longest distance strategies on the robustness tend to be overlapping, because the selected nodes are similar to each other in some cases. To distinguish the effects as much as possible, we introduce some link addition methods based on the two strategies. Through numerical simulations, we show that the robustness is the most improved by adding links on the minimum degree strategy for synthetic and real networks. As an exception, when the number of added links is small to an initial tree, the robustness is the most improved by adding links on the longest distance strategy. On the other hand, even combined with the minimum degree strategy, the robustness is only slightly improved by adding links on the shortest distance strategy for synthetic networks. Thus, adding links to pairs of the minimum degree nodes between longer distances is the most effective for improving robustness. However, when the minimum degree strategy selects nodes with longer distances on some real networks, the robustness is improved by adding links on both the minimum degree and shortest distance strategies. Moreover, we discuss the relation between improving the robustness and enhancing loops. From the obtained results, we emphasize that enhancing long loops is important for improving the robustness rather than short loops.

## Materials and methods

We explain the similarity of nodes selected by the minimum degree and longest distance strategies in considering a tree with many leaf nodes as shown in Fig 1A. The leaf nodes are the end of trees and have the minimum degree. The longest distance strategy selects nodes with the maximum distance, while these nodes are leaf nodes with the minimum degree. Conversely, the minimum degree strategy selects leaf nodes with the minimum degree. Many leaf nodes are located far away rather than close together. Thus, the minimum degree strategy tends to select nodes with longer distances. Since these strategies select similar nodes on a tree, the effects on the robustness are very overlapping. However, the similarity becomes weak on a network with loops. Fig 1B shows an extreme example in which these strategies select different nodes. In the network, the longest distance strategy selects nodes with the maximum degree, while the minimum degree strategy selects nodes with a short distance. First of all, we set a tree

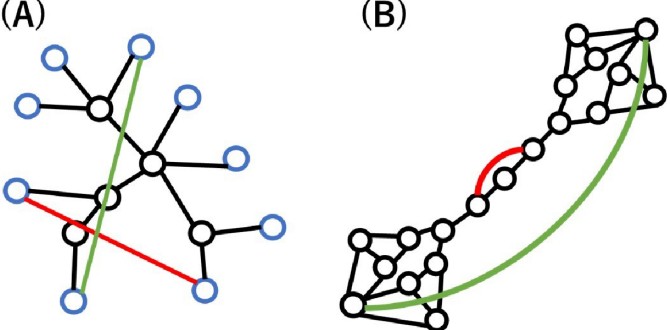

**Fig 1. Added links on the minimum degree and longest distance strategies.** (A) The minimum degree and longest distance strategies select similar nodes on a tree, while (B) these strategies select different nodes on a network with loops. Red and green links show added links on the minimum degree and longest distance strategies. Blue nodes are leaf nodes.

**Table 1. Six link addition methods in the combinations of the first step by degree and the second step by distance.**

| First step candidates are | all nodes | selected by the minimum degree |
|---|---|---|
| Random distance | random additions method (RandAdd method) [16, 17] | min-$k$ random additions method (min-$k$ RandAdd method) [16, 18–20] |
| Longest distance | longest additions method (LongAdd method) [14] | min-$k$ longest additions method (min-$k$ LongAdd method) |
| Shortest distance | shortest additions method (ShortAdd method) | min-$k$ shortest additions method (min-$k$ ShortAdd method) |

as the initial network for adding links, and then set a real network to investigate the effects of strategies in tree and network with loops.

We introduce link addition methods with two-step selections for investigating the effects of the minimum degree and longest distance strategies on the robustness against attacks. At the first step for linking, we consider two types: candidates are all nodes or selected from nodes with the minimum degree. Here, we omit to select nodes with the maximum degree, because it does not obviously contribute to improving the robustness. At the second step, we consider three types: a pair of nodes is selected uniformly at random, by the longest, or by the shortest distances in the candidates. In repeating these two steps, we add links to the selected pairs in the combinations as shown in Table 1. In Table 1, the minimum degree strategy is applied to the min-$k$ RandAdd, min-$k$ LongAadd, and min-$k$ ShortAdd methods. The longest distance strategy is applied to the LongAdd and min-$k$ LongAdd methods. In addition, we consider the shortest distance strategy for distinguishing the effects of long and short loops. The shortest distance strategy is applied to the ShortAdd and min-$k$ ShortAdd methods. The RandAdd method is the basic random additions. By applying the six link addition methods, we investigate the effects of the minimum degree and longest strategies on the robustness.

In general, as increasing the number of links, the robustness is more improved. For example, the complete graph with $N$ nodes and the maximum $N(N-1)/2$ links has the optimal robustness against any attacks (and failures). Thus, we consider effective link addition methods to improve the robustness by adding $\mathcal{O}(N)$ links. We apply the above link addition methods to synthetic and real-world networks for investigating the effects of these strategies on the robustness.

We first introduce synthetic networks and then explain real networks in the later paragraph. As synthetic networks, we consider random trees. Since trees are very fragile against attacks, we can clearly show the effects of improvement by adding a small number of links. In particular, because degree distribution strongly affects the robustness [2], we consider random trees with degree distributions ranging from power-law to exponential or narrower distributions. For generating random trees with the different degree distributions, we use a growing network (GN) model [24–26] and an inverse preferential attachment (IPA) model [27], which are extensions of the Barabási-Albert model [1]. In these models, a new node is added and connects a link to the existing node at each time step. By repeated attachments from the initial configuration of two connected nodes, the network becomes a tree with $N$ nodes and $N-1$ links. The degree distributions vary according to a value of parameter $v$ or $\beta$ as follows. In the GN model, the connection probability from a new node to the existing node $i$ is proportional to $k_i^v$, $v \geq 0$, where $k_i$ denotes the degree of node $i$. The degree distribution is a power-law distribution for $v = 1$, a power-law distribution with an exponential cut-off for $0 < v < 1$, and an exponential distribution for $v = 0$ [1]. In the IPA model, the connection probability is proportional to $k_i^{-\beta}$, $\beta \geq 0$. As $\beta$ becomes larger, the degree distribution changes from exponential ($\beta$

**Table 2. Properties of real networks.**

| Name | | $N$ | $M$ | $r$ |
|---|---|---|---|---|
| AirTraffic | [30] | 1226 | 2408 | -0.015 |
| Email | [31] | 1133 | 5451 | 0.078 |
| Yeast | [32] | 2224 | 6594 | -0.11 |

= 0) to narrower distributions. Note that the GN model with $v = 0$ and the IPA model with $\beta = 0$ are the same as uniformly random attachments. By using the GN and IPA models, we can generate random trees with degree distributions ranging from power-law to exponential or narrower distributions. Moreover, we also use configuration models [28, 29] for the random trees, because networks generated by the IPA model tend to have special chain-like structures [27]. In the configuration models without chain-like structures, it is numerically found that the robustness increases as the degree distributions become narrower [15]. Similar results are obtained for configuration models (See S1–S5 Figs).

As real networks, we consider an airline network [30], e-mail interchanges [31], and protein-protein interactions in budding yeast [32]. These networks are categorized as technological, social, and biological networks. Table 2 shows the number $N$ of nodes, the number $M$ of links, and the degree-degree correlation $r$ [6] for these networks.

For the link addition methods to synthetic and real-world networks, we compare the robustness index against malicious attacks [3]. As malicious attacks, we consider two types of attacks: the typical high degree adaptive (HDA) attacks [3] and the state-of-the-art most destructive belief propagation (BP) attacks [10]. The HDA attacks remove nodes in increasing order of degrees with recalculating degrees, while the BP attacks destroy loops by removing nodes with the highest probability of belonging to the FVS. For two types of attacks, we measure the robustness index $R = 1/N \sum_{q=1/N}^{1} S(q)$, where $S(q)$ denotes the number of nodes in the largest connected components after $qN$ nodes are removed [3]. The values of $R$ range from 0 to 0.5. In the distinction by suffixes, $R_{HDA}$ and $R_{BP}$ denote the robustness index against the HDA and BP attacks, respectively. We also compare the size of FVS. However, it is difficult to obtain the exact solution of the minimum FVS for a large network, because to find the minimum FVS is NP-hard in combinatorial optimization problems. Therefore, we apply an approximate method by message-passing based on statistical physics [33].

As another measure of network topology that differs from the robustness index, we consider network efficiency. There is a trade-off between robustness and network efficiency in SF networks [34], whereas high robustness coexists with network efficiency in incrementally growing onion-like networks [35]. In link addition methods, several previous methods have been proposed for improving only the efficiency [20, 36]. Therefore, we investigate the impact of the minimum degree and longest distance strategies on not only the robustness but also network efficiency $E$ and diameter $D$ defined as the maximum length of shortest paths in the network. Here, network efficiency $E$ is given by $E = 1/(N(N-1))\sum_{ij} 1/d_{ij}$, where $d_{ij}$ denotes the shortest path length counted by the hops between nodes $i$ and $j$ [37].

## Results

We investigate the robustness against the typical HDA and the worst-case BP attacks in comparing the corresponding size of FVS by applying six link addition methods. First, we show the results on random trees and SF networks as synthetic networks. Next, we show the results on real networks. As random trees, we set three random trees generated by the IPA model with $\beta$

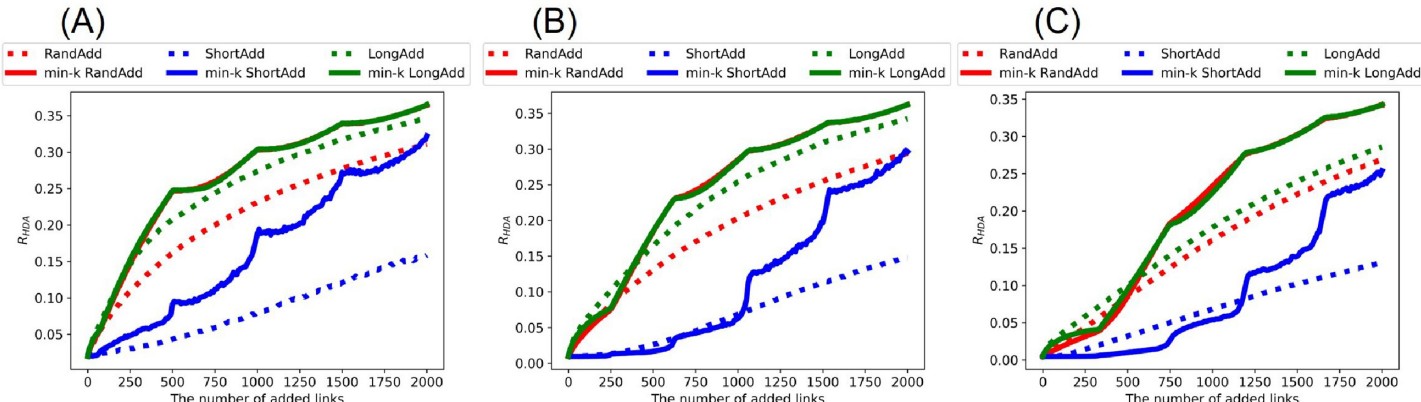

**Fig 2. Robustness index against typical HDA attacks after adding links to random trees.** The min-$k$ RandAdd and min-$k$ LongAdd methods (red and green solid lines) are the most effective for improving $R_{HDA}$. For small numbers of added links less than (A) 170, (B) 380, and (C) 530, the LongAdd method (green dotted lines) is higher than other five methods. The initial random tree has (A) narrower, (B) exponential, and (C) power-law degree distributions. Red, blue, and green dotted lines correspond to RandAdd, ShortAdd, and LongAdd methods. Red, blue, and green solid lines correspond to min-$k$ RandAdd, min-$k$ ShortAdd, min-$k$ LongAdd methods.

= 5 or the GN model with $v = 0$ or 1. The random trees have degree distributions following narrower, exponential, or power-law distributions. The following results are averaged over 100 samples.

Fig 2A–2C show $R_{HDA}$ after adding links to the initial random trees with narrower, exponential, and power-law degree distributions. It is trivial that $R_{HDA}$ is increased, as the number of added links increases. Thus, we compare the slopes of lines corresponding to these methods in increasing $R_{HDA}$. Fig 2A–2C show that red and green solid lines (denote min-$k$ RandAdd and min-$k$ LongAdd methods) have the largest slopes among the six lines. Moreover, the difference between red and green solid lines is very small. From these results, both min-$k$ RandAdd and min-$k$ LongAdd methods are the most effective for improving $R_{HDA}$ on a tree. Therefore, the minimum degree strategy contributes to improving the robustness. As an exception, green dotted line (denotes LongAdd method) is higher than the other five lines, when the number of added links is less than 170, 380, and 530 in Fig 2A–2C, respectively. Thus, only when the number of added links is small to an initial tree, the longest distance strategy is effective for improving the robustness. On the other hand, blue dotted and solid lines (denote ShortAdd and min-$k$ ShortAdd methods) have smaller slopes in Fig 2A–2C. In particular, blue dotted lines are lower than the other five lines. Therefore, the contribution is small to improving the robustness from initial random trees on the shortest distance strategy.

Fig 3 shows $R_{BP}$ after adding links to the random trees. Although the values of $R_{BP}$ are slightly smaller than the values of $R_{HDA}$, the whole trend is almost same as shown in Fig 2. Once again, the min-$k$ RandAdd and min-$k$ LongAdd methods are the most effective for improving $R_{BP}$. As an exception, the LongAdd method is the most effective for improving $R_{BP}$, only when the number of added links is small in the range of less than 130, 380, and 540 added links in Fig 3A–3C, respectively. On the other hand, the ShortAdd method has the smallest slope among six methods in increasing $R_{BP}$.

Fig 4A–4C show the size of FVS after adding links to the initial random trees. Similar to the behavior of $R_{HDA}$, it is trivial that as the number of added links increases, the size of FVS is increased. Thus, we compare the slopes of lines in increasing the size of FVS. Fig 4A–4C show that red and green solid lines (denote min-$k$ RandAdd and min-$k$ LongAdd method) have larger slopes than the other three lines except for the blue solid line (denotes min-$k$ ShortAdd

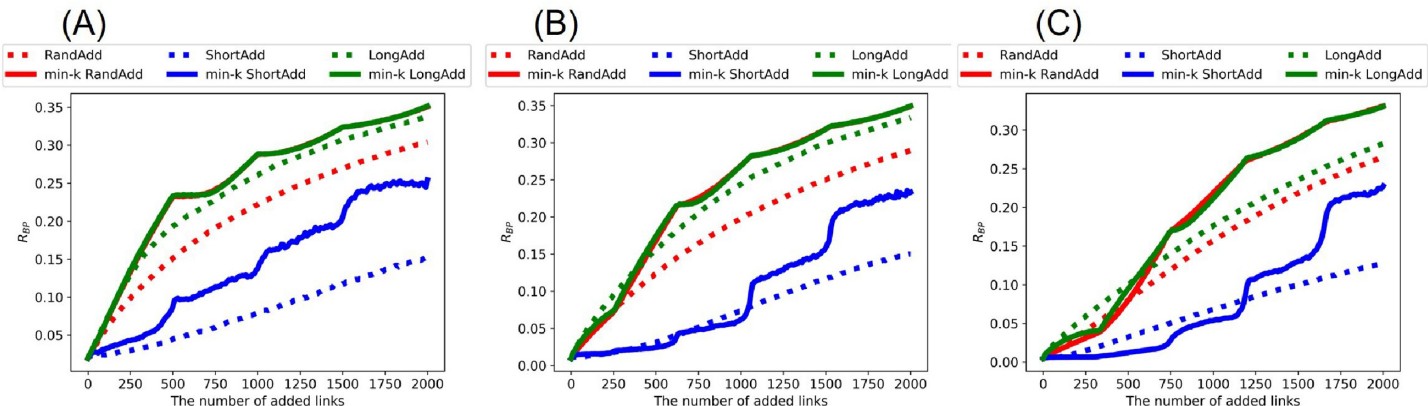

**Fig 3. Robustness index against the worst-case BP attacks after adding links to random trees.** The min-$k$ RandAdd and min-$k$ LongAdd methods (red and green solid lines) are the most effective for improving $R_{BP}$. For small numbers of added links less than (A) 130, (B) 380, and (C) 540, the LongAdd method (green dotted lines) is higher than other five methods. The initial random tree has (A) narrower, (B) exponential, and (C) power-law degree distributions. The correspondence between lines and methods is the same as in Fig 2.

method). Red and green solid lines are higher than the three lines in the range of more than 780, 460, and 890 added links in Fig 4A–4C, respectively. We remark that red and green solid lines outperform the other lines for increasing $R_{HDA}$ and $R_{BP}$ in Figs 2 and 3. Thus, the min-$k$ RandAdd and min-$k$ LongAdd methods are effective for both improving the robustness and increasing the size of FVS. Conversely, as shown in the blue solid line, the effect of the min-$k$ ShortAdd method is strong for increasing the size of FVS but weak for improving the robustness. Blue solid line has the largest slope among the six lines for increasing the size of FVS in Fig 4, although this line has a small slope in increasing $R_{HDA}$ as shown Fig 2. The reason of a large slope in increasing the size of FVS is considered as that the min-$k$ ShortAdd method generates many short loops. By the min-$k$ ShortAdd method, triangles are generated between leaf nodes with a same neighbor node. Such triangles increase the size of FVS but do not contribute to improving the robustness.

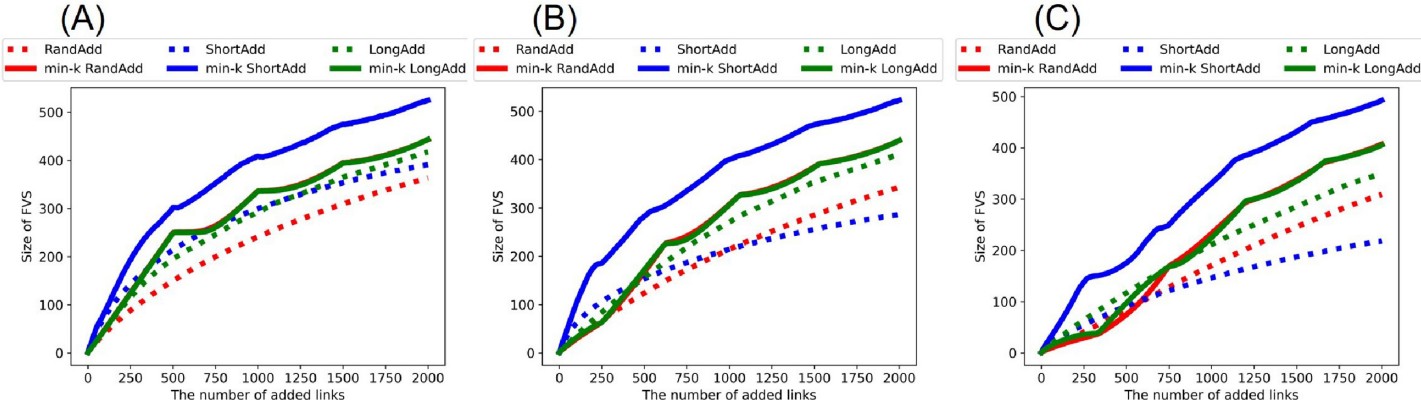

**Fig 4. Size of FVS after adding links to random trees.** The min-$k$ RandAdd and min-$k$ LongAdd methods (red and green solid lines) are effective for increasing the size of FVS. In each figure of(A)(B)(C), the min-$k$ ShortAdd method (a blue line) outperforms the other methods. The initial random tree has (A) narrower, (B) exponential, and (C) power-law degree distributions. The correspondence between lines and methods is same as shown in Fig 2.

**Table 3. The most effective methods for improving $R_{\text{HDA}}$, $R_{\text{BP}}$, and the size of FVS to random trees.**

|  |  | $\beta = 5$ | $\nu = 0$ | $\nu = 1$ |
|---|---|---|---|---|
| $R_{\text{HDA}}$ | Fig 2 | min-$k$ RandAdd | min-$k$ RandAdd | min-$k$ RandAdd |
|  |  | min-$k$ LongAdd | min-$k$ LongAdd | min-$k$ LongAdd |
| $R_{\text{BP}}$ | Fig 3 | min-$k$ RandAdd | min-$k$ RandAdd | min-$k$ RandAdd |
|  |  | min-$k$ LongAdd | min-$k$ LongAdd | min-$k$ LongAdd |
| Size of FVS | Fig 4 | min-$k$ ShortAdd | min-$k$ ShortAdd | min-$k$ ShortAdd |

Table 3 summarizes the results for improving the robustness and the size of FVS from initial random trees. The min-$k$ RandAdd and min-$k$ LongAdd methods are the most effective for increasing both $R_{\text{HDA}}$ and $R_{\text{BP}}$ from ones in random trees. On the other hand, the effect of the min-$k$ Short method is strong for increasing the size of FVS but weak for increasing the robustness. From these results, the minimum degree strategy contributes significantly to improving the robustness from initial random trees. However, even on the minimum degree strategy, adding links to pairs of nodes with short distances only slightly improves the robustness.

In SF networks, we investigate the effects of node size $N$ and average degree $\langle k \rangle$ on the improvement of robustness by link addition methods. The SF networks are generated by the GN model with $\nu = 1$, which is the same as Barabási-Albert model. Fig 5 shows $R_{\text{HDA}}$ after adding links on SF networks with $N = 500$, $1000$, and $5000$ and $\langle k \rangle = 2$ and $4$. Note that the SF network with $\langle k \rangle = 2$ is a tree. Fig 5 shows similar results even for different node sizes $N$. In comparing the ratios of added and existing links, the trend of improving the robustness of each strategy is almost the same. The trend means that, as shown in Fig 5, the orders of lines from top to bottom are similar at the same ratio of added links for $N = 500$, $1000$, and $5000$. These results show that the effect of node size is negligible. When the number of added links is 10 times the number of existing links, the five lines except the blue dotted line (ShortAdd method) have almost similar slopes, as shown in Fig 5A for $N = 500$. For sufficiently large numbers of added links, the ratio of added links strongly affects the improvement of the robustness more than the differences between methods. As the impact of the initial average degree $\langle k \rangle$ on the improvement of robustness, Fig 5A–5C for $\langle k \rangle = 2$ show that blue solid line (denotes min-$k$ ShortAdd method) has smaller slope than red and green solid lines (denote min-$k$ RandAdd and min-$k$ LongAdd methods), whereas Fig 5D–5F for $\langle k \rangle = 4$ show that the blue solid line has a similar or larger slope than red and green solid lines. From these results, the min-$k$ ShortAdd method is effective for non-tree SF networks. The reason is considered as follows. For non-tree networks, such triangles are not generated, since there are no leaf nodes. However, the min-$k$ ShortAdd method is ineffective for trees. For trees, adding links by the min-$k$ ShortAdd method generates triangles between leaf nodes that is not related to improve the robustness. Thus, it is considered that the min-$k$ ShortAdd method improves the robustness effective by making the degree distribution homogeneous without constructing local triangles. These results are supported from that the effect of the minimum degree strategy becomes greater as the average degree $\langle k \rangle$ increases and networks become further different from trees.

Since it is found that the homogeneous degree distribution strongly affects the improvement of the robustness, we investigate the changes of degree distributions by link addition methods. Fig 6 show the variances of degree distributions after adding links to SF networks with $N = 1000$ and $\langle k \rangle = 2$ and $4$. For the min-$k$ RandAdd, min-$k$ ShortAdd, and min-$k$ LongAdd methods, variances are the same value, since they select the minimum degree nodes. In

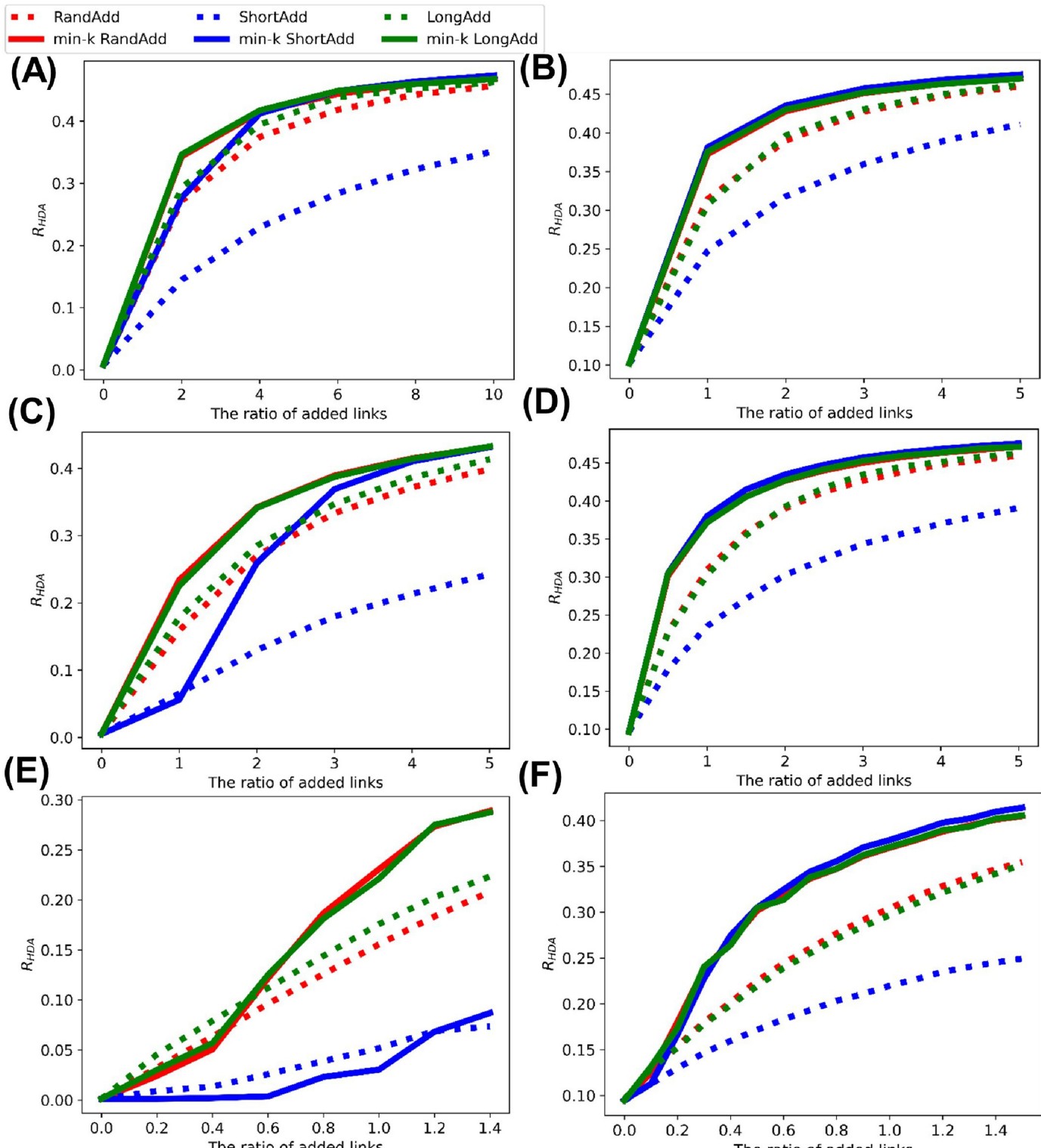

**Fig 5. Robustness index against typical HDA attacks after adding links to SF networks.** The min-*k* RandAdd and min-*k* LongAdd methods (red and green solid lines) are effective for SF trees, while the min-*k* RandAdd, min-*k* ShortAdd, and min-*k* LongAdd methods (three solid lines) are effective for non-tree SF networks. The initial SF networks are trees with (A) $N = 500$, (C) 1000, or (E) 5000, and are non-trees with (B) $N = 500$, (D) 1000, or (F) 5000 nodes.

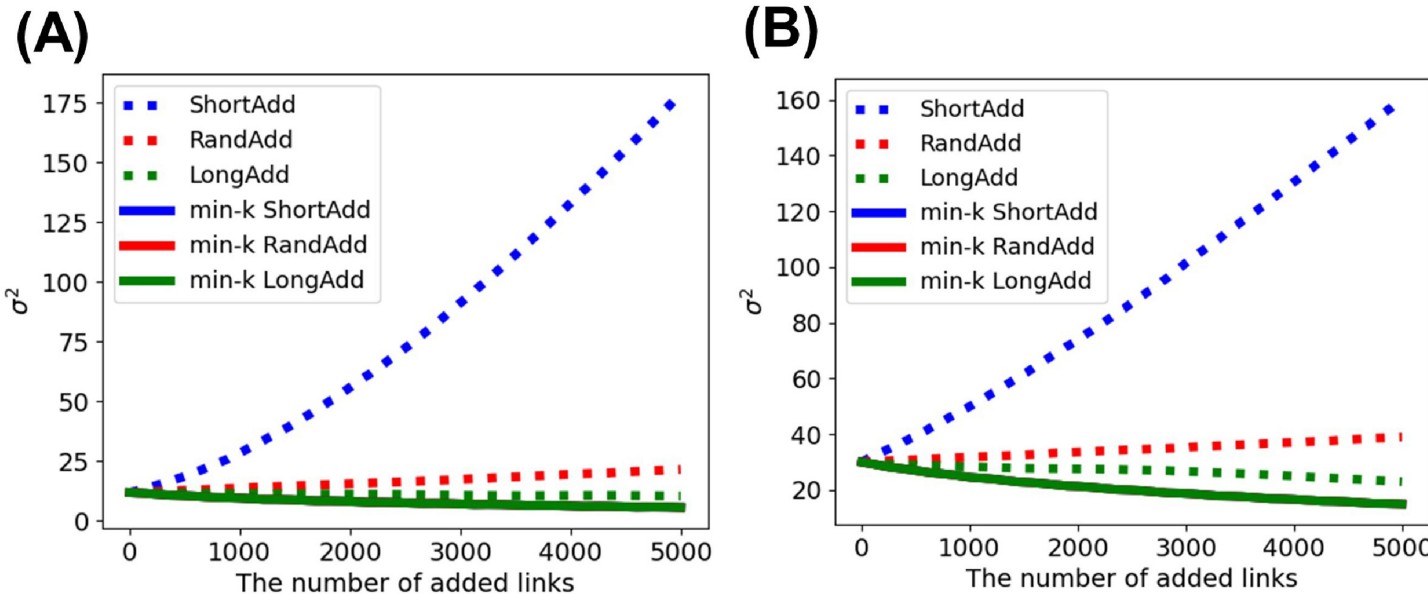

**Fig 6. Variances of degree distributions after adding links to SF networks.** The LongAdd method (green dotted line) decreases the variance, while the ShortAdd method (blue dotted line) increases the variance. The min-$k$ RandAdd, min-$k$ ShortAdd, and min-$k$ LongAdd methods (three solid lines) are the same. The initial networks are SF networks with $N = 1000$ and (A) $\langle k \rangle = 2$ or (B) 4.

the other three methods, the LongAdd method decreases variance, while the ShortAdd method increases the variance. This means that the longest distance strategy makes the degree distribution homogeneous. Fig 7 shows the degree distributions after adding links to SF networks. In the min-$k$ RandAdd, min-$k$ ShortAdd, and min-$k$ LongAdd methods, lower degrees are rightly shifted with unchanged higher degrees. Similar changes can be seen for the LongAdd method. Compared to degree distributions in the RandAdd methods, lower degrees are rightly shifted with unchanged higher degrees in the LongAdd method. The longest degree strategy tends to add links to nodes with lower degrees and makes the degree distribution homogeneous. On the other hand, in the ShortAdd method, lower degrees change little, and medium degrees are rightly shift. The shortest distance strategy tends to add nodes with higher degrees to increase the variance of degree distribution.

We discuss that the distance between selected nodes for adding links, since there is no difference in degree distributions for three min-$k$ methods. Fig 8 show the distance between selected nodes for adding links to SF networks with $N = 1000$ and $\langle k \rangle = 2$ and 4. It is obvious that the LongAdd method selects node pairs with the longest distance and the ShortAdd method selects node pairs with the shortest distance. The RandAdd method selects node pairs with about middle distance nodes between the LongAdd and ShortAdd methods. For three min-$k$ methods, the min-$k$ LongAdd method tends to select node pairs with longer distance and the min-$k$ ShortAdd method tends to select node pairs with shorter distance. However, the LongAdd method selects node pairs with longer than the min-$k$ LongAdd methods, which select node pairs from nodes with the minimum degree. Moreover, for the min-$k$ ShortAdd method and min-$k$ LongAdd method, the distances between selected node pairs sharply change when the value of the minimum degree changes.

In summary, we investigate that node size has a small effect, but the average degree has a strong effect on the improvement of the robustness. In particular, the min-$k$ ShortAdd method is effective for improving the robustness for non-tree SF networks, while it is ineffective for

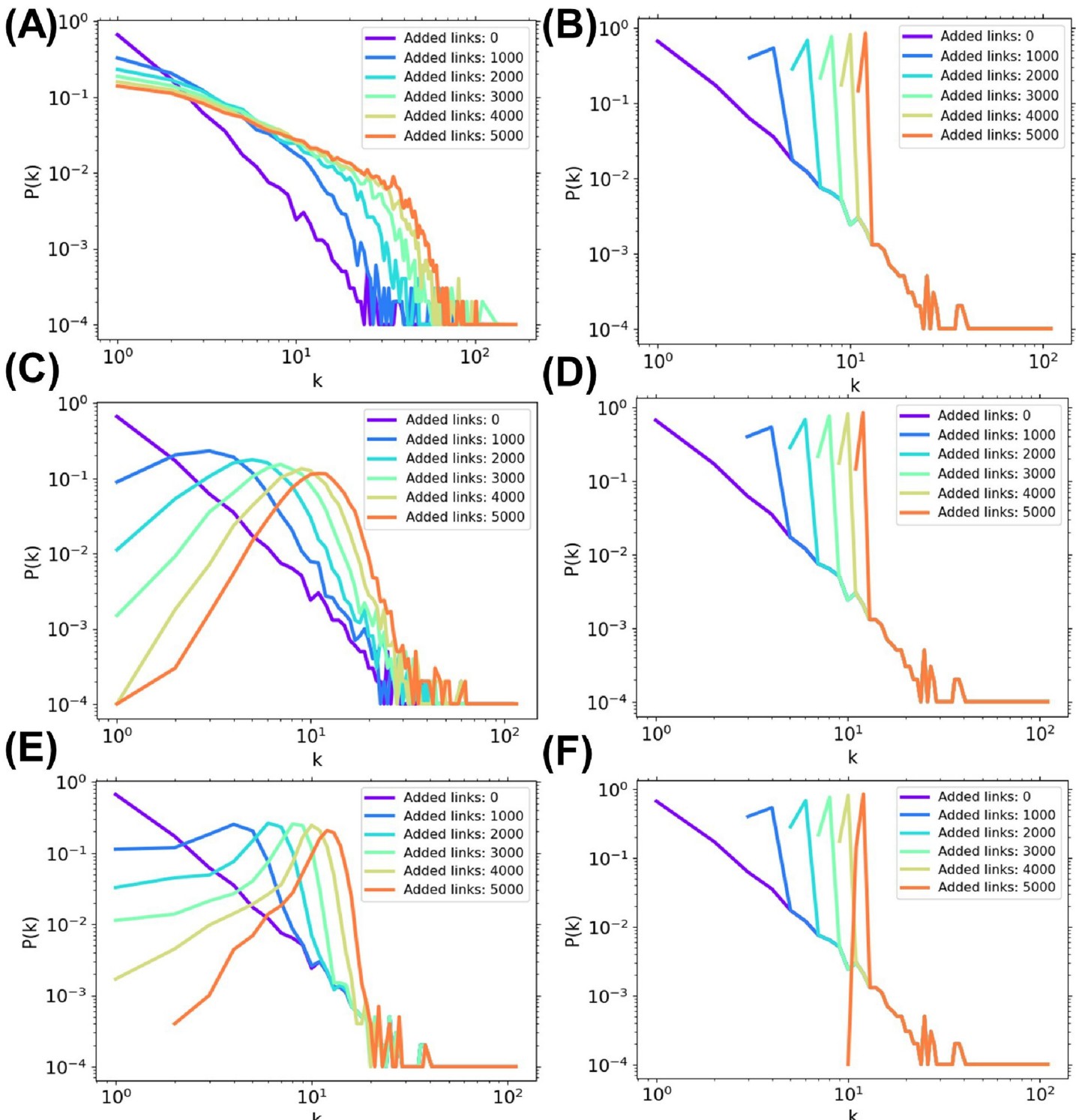

**Fig 7. Degree distributions after adding links to SF networks.** The LongAdd method adds links to nodes with lower degrees, while the ShortAdd method adds links to nodes with middle to higher degrees. The results are shown for by (A) ShortAdd, (B) min-$k$ ShortAdd, (C) RandAdd, (D) min-$k$ RandAdd, (E) LongAdd, and (F) min-$k$ LongAdd methods, respectively. The initial networks are SF networks with $N = 1000$ and $\langle k \rangle = 2$.

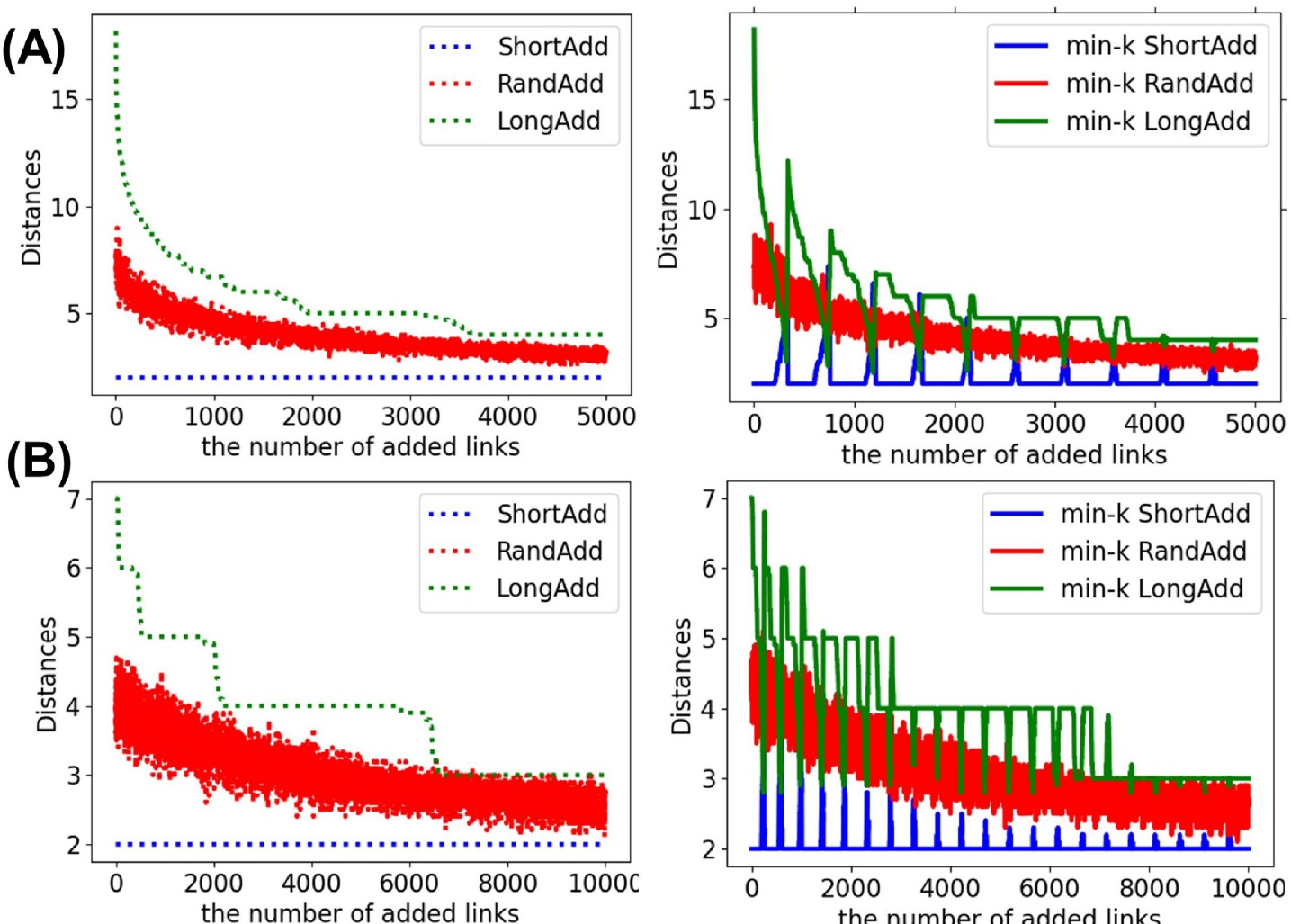

**Fig 8. Distance between selected node pairs for adding links to SF networks.** The LongAdd method (green dotted lines) selects node pairs with the longer distances, while the ShortAdd method (blue dotted lines) selects node pairs with the shorter distances. The LongAdd method (green dotted lines) selects node pairs with longer than the min-$k$ LongAdd methods (green solid lines). The initial networks are SF networks with $N = 1000$ and (A) $\langle k \rangle = 2$ or (B) 4.

trees. There is no difference between the three min-$k$ methods in improving the robustness for non-tree SF networks, although these methods select nodes with different distances. Therefore, for non-tree SF networks, it is suggested that the effect of adding links to the minimum degree nodes and that homogeneous degree distribution is more important for improving the robustness than the effect of distance.

Next, we explain the results on real networks. Fig 9A–9C show $R_{HDA}$ after adding links to initial real networks: AirTraffic, Email, and Yeast. Similar to the results of $R_{HDA}$ on random trees in Fig 2, the min-$k$ RandAdd and min-$k$ LongAdd methods are the most effective for improving $R_{HDA}$. Unlike the results on random trees, the min-$k$ ShortAdd method is the best on real networks. Fig 9A–9C show that the slope of blue solid line (denotes min-$k$ ShortAdd method) is close to those of red and green solid lines, as the number of added links increases. From these results, the minimum degree strategy significantly contributes to improving the robustness on real networks. In other words, by comparing to the minimum degree strategy,

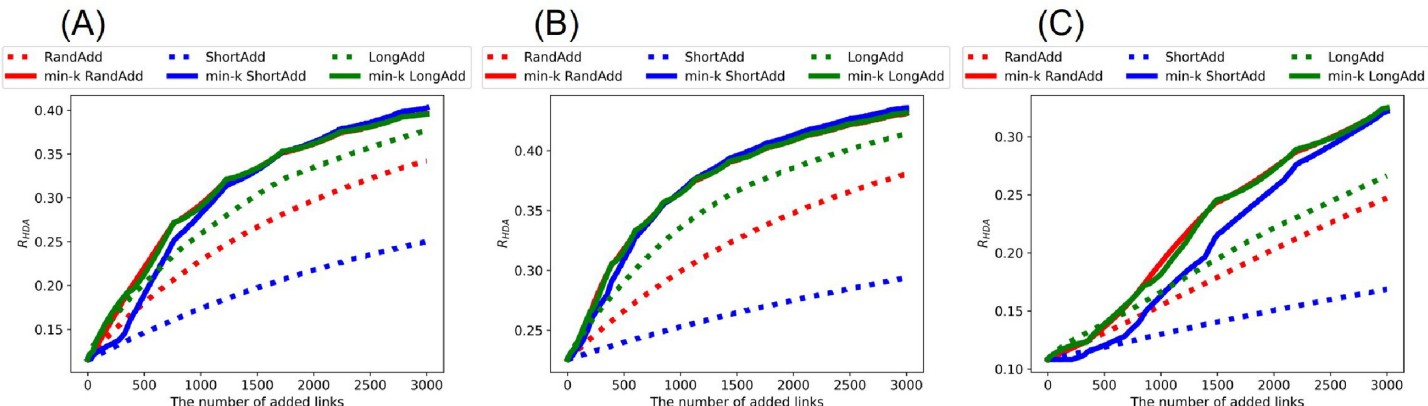

**Fig 9. Robustness index against typical HDA attacks after adding links to real networks.** The min-*k* RandAdd, min-*k* LongAdd, and min-*k* ShortAdd methods (red, green, and blue solid lines) are the most effective for improving $R_{HDA}$. The initial networks are (A) AirTraffic, (B) Email, and (C) Yeast. The correspondence between lines and methods is the same as shown in Fig 2.

the effect of the longest distance strategy is weaker, especially when the number of added links is large. This is different from the result on the tree. We later discuss the strong effect of the minimum degree strategy on the distance between nodes in real networks. It is considered that the minimum degree strategy selects pairs of nodes with sufficient longer distances for real networks even on the minimum distance strategy.

Fig 10A–10C show $R_{BP}$ after adding links to the real networks. The whole trend is almost same as shown in Fig 9. Once again, the min-*k* RandAdd, min-*k* LongAdd, and min-*k* ShortAdd methods are the most effective for improving $R_{BP}$. In particular, as the number of added links increases, the min-*k* ShortAdd method becomes more effective. Similarly, Fig 11A–11C show the size of FVS after adding links to the real networks. As similar to results of the size of FVS on random trees in Fig 4, the min-*k* ShortAdd method is the most effective for increasing the size of FVS. Blue solid line is higher than the other five lines in Fig 11A–11C. Thus, on the real networks, the min-*k* ShortAdd method is effective for increasing both the robustness and the size of FVS.

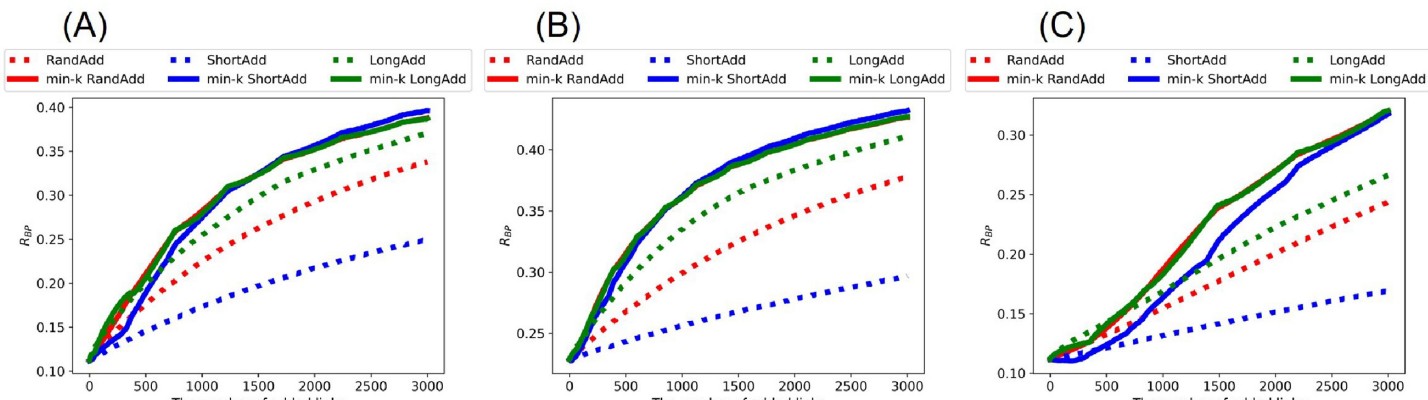

**Fig 10. Robustness index against the worst-case BP attacks after adding links to real networks.** The min-*k* RandAdd, min-*k* LongAdd, and min-*k* ShortAdd methods (red, green, and blue solid lines) are the most effective for improving the robustness. The initial networks are (A) AirTraffic, (B) Email, and (C) Yeast. The correspondence between lines and methods is the same as shown in Fig 2.

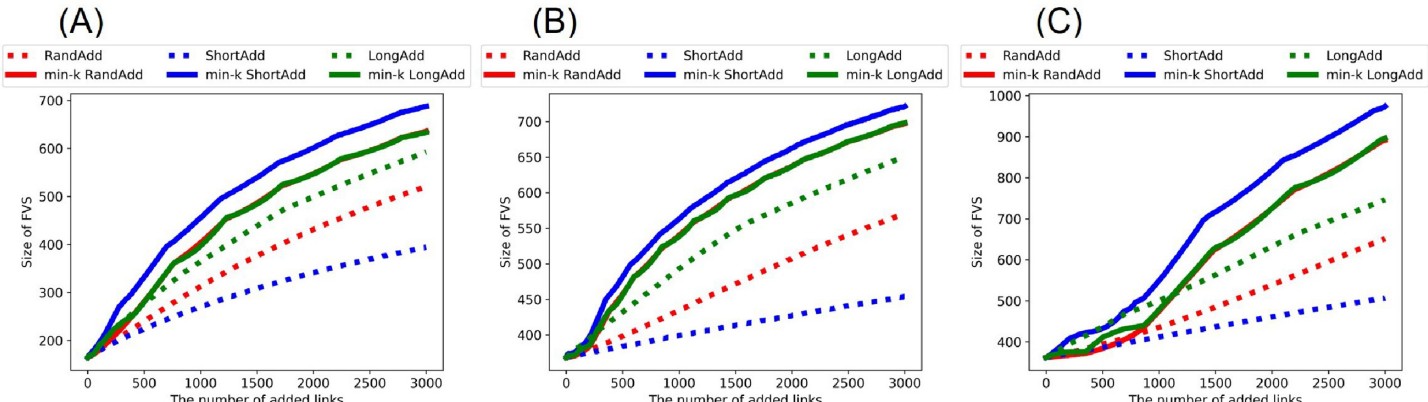

**Fig 11. Size of FVS after adding links to real networks.** In each figure of(A)(B)(C), the min-*k* ShortAdd method (a blue line) outperforms the other methods. The min-*k* RandAdd and min-*k* LongAdd methods (red and green solid lines) are also effective for increasing the size of FVS. The initial networks are (A) AirTraffic, (B) Email, and (C) Yeast. The correspondence between lines and methods is the same as shown in Fig 2.

Table 4 summarizes the results for improving the robustness and the size of FVS from initial real networks. The min-*k* RandAdd, min-*k* LongAdd, and min-*k* ShortAdd methods are the most effective for increasing both $R_{HDA}$ and $R_{BP}$ for AirTraffic and Email. For Yeast, the min-*k* RandAdd and min-*k* LongAdd methods are the most effective. As a larger number of added links, the min-*k* ShortAdd method becomes more effective for all three networks. From these results, the contribution is stronger on the minimum degree strategy to improving the robustness, as the number of added links increases. Conversely, the contribution is weaker on the longest distance strategy.

For investigating the effects of link addition methods on the distances between nodes, we show the changes of the topological structure measured by diameter *D* and network efficiency *E*. Remember that there is a trade-off between robustness and network efficiency in scale-free networks [34], while high robustness coexists with network efficiency in incrementally growing onion-like networks [35]. Thus, it is not trivial that the distances become smaller when the robustness is strong. In general, as the number of added links increases, *D* decreases while *E* increases. Thus, we compare the slopes of lines in decreasing *D* or increasing *E*. Fig 12A–12C show the diameter *D* after adding links to the initial random trees with narrower, exponential, and power-law degree distributions. Fig 12A–12C show that green dotted line (denotes Long-Add method) has the largest slope in decreasing *D* among six lines. Since *D* is the maximum length in all shortest paths, it is clear that *D* is the most decreased by adding the longest links. Moreover, the slope of red solid line (denotes min-*k* RandAdd methods) is close to that of

**Table 4. The most effective methods for improving $R_{HDA}$, $R_{BP}$, and the size of FVS to real networks.**

|  |  | AirTraffic | Email | Yeast |
|---|---|---|---|---|
| $R_{HDA}$ | Fig 9 | min-*k* RandAdd | min-*k* RandAdd | min-*k* RandAdd |
|  |  | min-*k* LongAdd | min-*k* LongAdd | min-*k* LongAdd |
|  |  | min-*k* ShortAdd | min-*k* ShortAdd |  |
| $R_{BP}$ | Fig 10 | min-*k* RandAdd | min-*k* RandAdd | min-*k* RandAdd |
|  |  | min-*k* LongAdd | min-*k* LongAdd | min-*k* LongAdd |
|  |  | min-*k* ShortAdd | min-*k* ShortAdd |  |
| Size of FVS | Fig 11 | min-*k* ShortAdd | min-*k* ShortAdd | min-*k* ShortAdd |

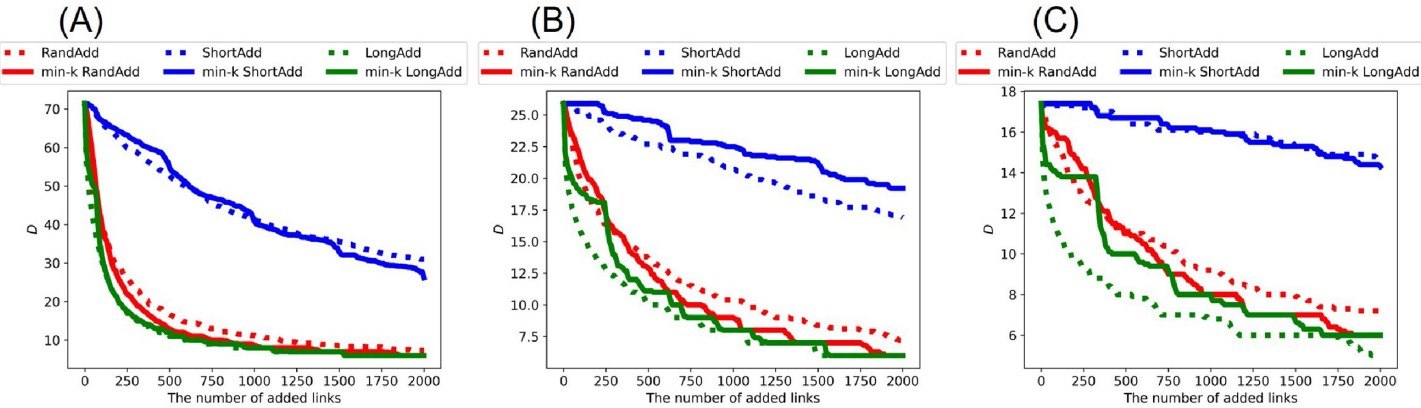

**Fig 12. Diameter after adding links to random trees.** The LongAdd method (green dotted line) is the most effective for decreasing $D$ among the six methods. The ShortAdd and min-$k$ ShortAdd method (blue dotted and solid lines) are higher than other four methods. The initial random tree has (A) narrower, (B) exponential, and (C) power-law degree distributions. The correspondence between lines and methods is the same as in Fig 2.

green dotted line. Therefore, the minimum degree strategy contributes to decreasing $D$. On the other hand, blue solid line (denotes min-$k$ ShortAdd method) has a smaller slope in decreasing $D$ than the other four lines. The slope of blue solid line is close to that of blue dotted lines (denotes ShortAdd method). For initial random trees, the effect of the minimum shortest strategy is weak in decreasing $D$ even on the minimum degree strategy.

Fig 13A–13C show the diameter $D$ after adding links to initial real networks: AirTraffic, Email, and Yeast. Similar to the results on random trees in Fig 12, the longest distance strategy is the most effective for decreasing $D$ in real networks. Furthermore, the minimum degree strategy contributes to decreasing $D$. Unlike the results on random trees, the min-$k$ ShortAdd method is more effective for decreasing $D$ in AirTraffic and Email. In Fig 13A and 13B, the slope of blue solid line (denotes min-$k$ ShortAdd method) is close to that of red dotted line (denotes RandAdd method) rather than blue dotted line (denotes ShortAdd method). Thus, it is considered that the min-$k$ ShortAdd method selects pairs of nodes with sufficient distances for decreasing $D$ for AirTraffic and Email. We remark that the min-$k$ ShortAdd method is the

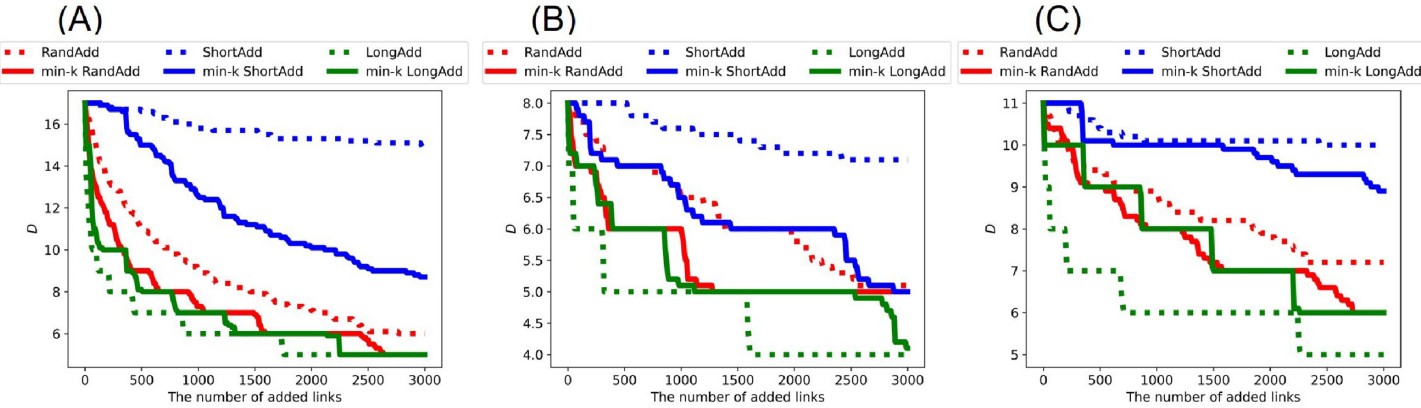

**Fig 13. Diameter after adding links to real networks.** The LongAdd method (green dotted line) is the most effective for decreasing $D$ among the six methods. The min-$k$ ShortAdd method (blue solid lines) becomes close to RandAdd method (red dotted line). The initial networks are (A) AirTraffic, (B) Email, and (C) Yeast. The correspondence between lines and methods is same as shown in Fig 2.

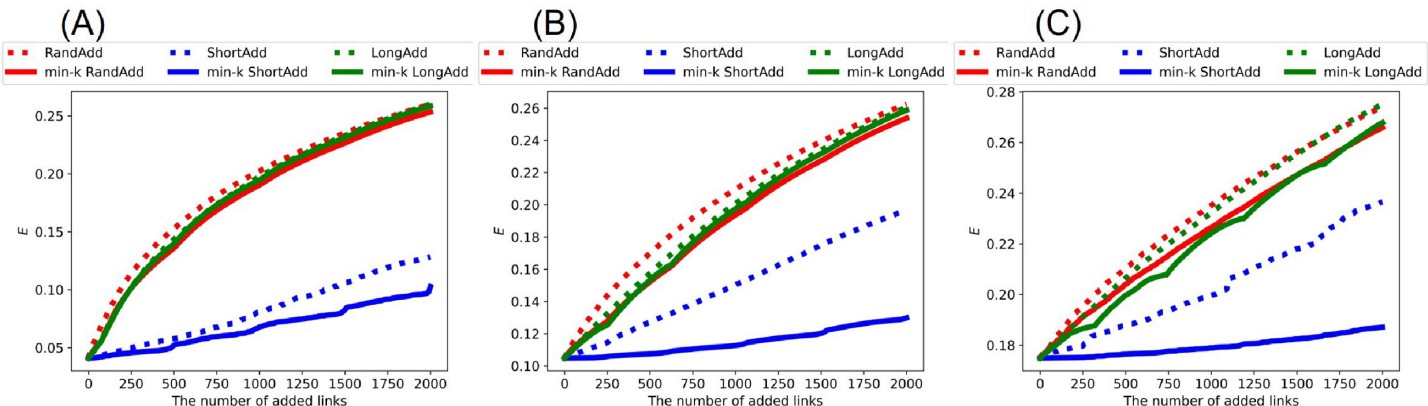

**Fig 14. Network efficiency after adding links to random trees.** The RandAdd method (red dotted line) is the most effective for increasing $E$ among six methods. The min-$k$ ShortAdd method (blue solid lines) is lower than the other five methods. The initial random tree has (A) narrower, (B) exponential, and (C) power-law degree distributions. The correspondence between lines and methods is the same as shown in Fig 2.

most effective for improving the robustness for AirTraffic and Email as shown in Table 4. On the other hand, the effect of min-$k$ ShotAdd method is a little weak in decreasing $D$ for Yeast. In Fig 13C, the slope of blue solid line is close to that of blue dotted line.

Similarly, Figs 14 and 15 show network efficiency $E$ after adding links to the random trees and real networks, respectively. We also compare the slopes of lines in increasing $E$. In general, the average distance between nodes becomes smaller, as $E$ increases. Fig 14A–14C show that red dotted line (denotes RandAdd method) is higher than the other five lines for initial random trees. Moreover, the slopes of green and red solid lines (denote min-$k$ RandAdd and min-$k$ LongAdd methods) are close to that of red dotted line. Thus, the min-$k$ RandAdd and min-$k$ LongAdd methods are effective for increasing both the robustness and $E$ for initial random trees. On the other hand, blue solid line (denotes min-$k$ ShortAdd method) is lower than the other five lines in Fig 14A–14C. As mentioned above, since the min-$k$ ShortAdd method generates triangles between leaf nodes on random trees, the contribution is small to increasing both $E$ and the robustness. Also for initial real networks, the RandAdd method is the most

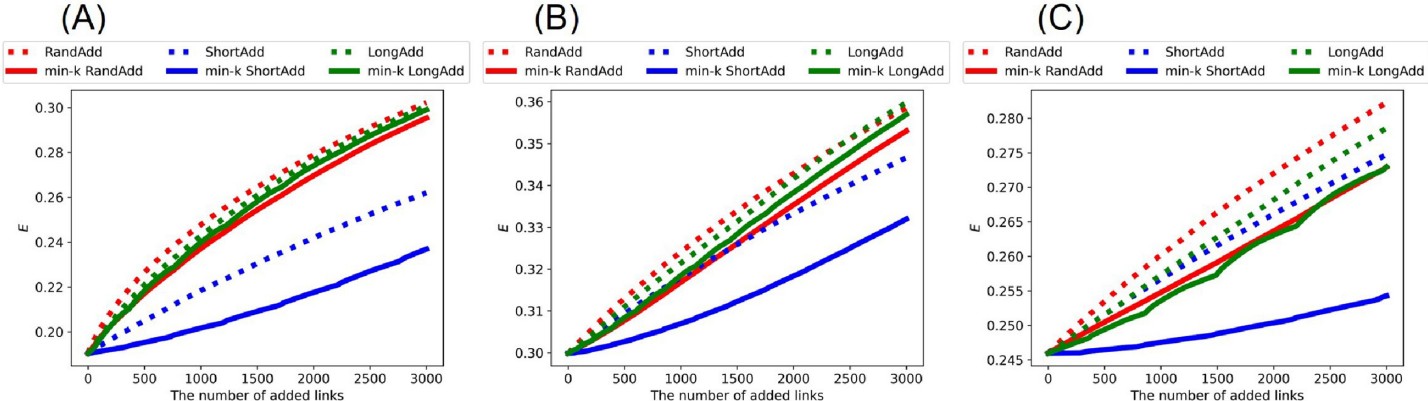

**Fig 15. Network efficiency after adding links to real networks.** The RandAdd method (red dotted line) is the most effective for increasing $E$ among six methods. The min-$k$ ShortAdd method (blue solid line) is lower than the other five methods. The initial networks are (A) AirTraffic, (B) Email, and (C) Yeast. The correspondence between lines and methods is the same as shown in Fig 2.

**Table 5. The most effective methods for decreasing *D* and increasing *E* to random trees and real networks.**

|  |  | $\beta = 5$ | $\nu = 0$ | $\nu = 1$ | AirTraffic | Email | Yeast |
|---|---|---|---|---|---|---|---|
| *D* | Figs 12 & 13 | LongAdd | LongAdd | LongAdd | LongAdd | LongAdd | LongAdd |
|  |  | min-*k* LongAdd | min-*k* LongAdd | min-*k* LongAdd | min-*k* LongAdd |  |  |
|  |  | min-*k* RandAdd | min-*k* RandAdd |  | min-*k* RandAdd |  |  |
|  |  | RandAdd |  |  |  |  |  |
| *E* | Figs 14 &15 | RandAdd | RandAdd | RandAdd | RandAdd | RandAdd | RandAdd |
|  |  | LongAdd | LongAdd | LongAdd | LongAdd | LongAdd |  |
|  |  | min-*k* LongAdd | min-*k* LongAdd |  | min-*k* LongAdd | min-*k* LongAdd |  |
|  |  | min-*k* RandAdd | min-*k* RandAdd |  | min-*k* RandAdd |  |  |

effective for increasing *E* as shown in Fig 15A–15C. Furthermore, the min-*k* RandAdd and min-*k* LongAdd methods are effective for increasing both the robustness and *E*, especially for AirTraffic and Email as shown in Fig 15A and 15B. For Yeast, *E* is slightly increased by the min-*k* RandAdd and min-*k* LongAdd methods as shown in Fig 15C. In contrast, although the min-*k* ShortAdd method is effective for improving the robustness, *E* is only slightly increased by the min-*k* ShortAdd method as shown in blue solid lines of Fig 15A–15C. Table 5 summarizes the most effective methods for decreasing *D* and increasing *E* from initial random trees and real networks.

## Conclusion

We consider the contributions of the minimum degree and longest distance strategies to improving the robustness by adding links. To distinguish the effects of degrees and distances as much as possible, we introduce link addition methods with two-step selections based on degrees or distances. It is expected that two strategies select similar nodes on a tree, while they select different nodes on a network with loops. Moreover, since the robustness is strongly affected by the degree distribution, we use initial random trees with different degree distributions following power-law, exponential, and narrower distributions generated by the GN [24–26] and IPA models [27]. Our results show that the min-*k* RandAdd and min-*k* LongAdd methods are the most effective for improving the robustness by adding links to an initial random tree. As an exception, only when the number of added links is small, the longest distance strategy is the best. Note that the min-*k* RandAdd and min-*k* LongAdd methods are also effective for increasing the size of FVS. On the other hand, the effect of min-*k* ShortAdd method is weak for improving the robustness by connections between leaves locally. Thus, it is suggested that enhancing long (global) loops is important for improving the robustness rather than short (local) loops for random trees. Similarly, for real networks, the min-*k* RandAdd, min-*k* LongAdd, and min-*k* ShortAdd methods are the most effective for improving the robustness. Even on the shortest distance strategy, the minimum degree strategy significantly contributes to improving the robustness on real networks. The reason may be considered as that the minimum degree strategy selects nodes with sufficient long distances in decreasing the diameter even on the shortest distance strategy. However, its detailed analysis requires further studies. In conclusion, adding links to pairs of the minimum degree nodes between longer distances is the most effective for improving the robustness on both synthetic trees and real networks. In other words, enhancing global loops is essential for improving the robustness and efficiency. For future studies, it may be useful to extend our methods of adding links for more complicated networks, such as multilayer networks [38], temporal networks [39], multi-fractal networks [40, 41], and brain connectomes [42, 43].

## Supporting information

**S1 Fig. Robustness index against typical HDA attacks after adding links to configuration models of random trees.** The min-$k$ RandAdd and min-$k$ LongAdd methods (red and green solid lines) are the most effective for improving $R_{HDA}$. For small numbers of added links less than (A) 170, (B) 380, and (C) 530, the LongAdd method (green dotted lines) is higher than other five methods. The initial network is configuration model of random tree with (A) narrower, (B) exponential, and (C) power-law degree distributions. Note that the network is randomly rewired for connecting all nodes.
(EPS)

**S2 Fig. Robustness index against the worst-case BP attacks after adding links to configuration models of random trees.** The min-$k$ RandAdd and min-$k$ LongAdd methods (red and green solid lines) are the most effective for improving $R_{BP}$. For small numbers of added links less than (A) 170, (B) 380, and (C) 530, the LongAdd method (green dotted lines) is higher than other five methods. The initial network is configuration model of random tree with (A) narrower, (B) exponential, and (C) power-law degree distributions. Note that the network is randomly rewired for connecting all nodes.
(EPS)

**S3 Fig. Size of FVS after adding links to configuration models of random trees.** The min-$k$ RandAdd and min-$k$ LongAdd methods (red and green solid lines) are effective for increasing the size of FVS. In each figure of(A)(B)(C), the min-$k$ ShortAdd method (a blue line) outperforms the other methods. The initial network is configuration model of random tree with (A) narrower, (B) exponential, and (C) power-law degree distributions. Note that the network is randomly rewired for connecting all nodes.
(EPS)

**S4 Fig. Diameter after adding links to configuration models of random trees.** The LongAdd method (green dotted line) is the most effective for decreasing $D$ among the six methods. The ShortAdd and min-$k$ ShortAdd method (blue dotted and solid lines) are higher than other four methods. The initial network is configuration model of random tree with (A) narrower, (B) exponential, and (C) power-law degree distributions. Note that the network is randomly rewired for connecting all nodes.
(EPS)

**S5 Fig. Network efficiency after adding links to configuration models of random trees.** The RandAdd method (red dotted line) is the most effective for increasing $E$ among six methods. The min-$k$ ShortAdd method (blue solid lines) is lower than the other five methods. The initial network is configuration model of random tree with (A) narrower, (B) exponential, and (C) power-law degree distributions. Note that the network is randomly rewired for connecting all nodes.
(EPS)

## Author Contributions

**Conceptualization:** Yukio Hayashi.

**Funding acquisition:** Yukio Hayashi.

**Investigation:** Masaki Chujyo, Yukio Hayashi.

**Methodology:** Masaki Chujyo, Yukio Hayashi.

**Supervision:** Yukio Hayashi.

**Visualization:** Masaki Chujyo.

**Writing – original draft:** Masaki Chujyo.

**Writing – review & editing:** Masaki Chujyo, Yukio Hayashi.

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
