## [Decision Letter · Decision Letter 0]

9 Aug 2022

PONE-D-22-09998Adding links on minimum degree and longest distance strategies for improving network robustness and efficiencyPLOS ONE

Dear Dr. Chujyo,

Thank you for submitting your manuscript to PLOS ONE. After careful consideration, we feel that it has merit but does not fully meet PLOS ONE’s publication criteria as it currently stands. Therefore, we invite you to submit a revised version of the manuscript that addresses the points raised during the review process. As you will see, the reviewers are in general favorable towards eventual publication of your work, but I agree with their comments that the manuscript can be substantially improved by a more careful discussion of recent literature and by providing more details on the merit of the work related to more realistic network topologies.

We look forward to receiving your revised manuscript.

Kind regards,

Lazaros K. Gallos

Academic Editor

PLOS ONE

Journal Requirements:

YH is partially supported by a Grant-in-Aid for Scientific Research (Grant Number JP.21H03425) from the Japan Society for the Promotion of Science. The funders had no role in study design, data collection and analysis, decision to publish, or preparation of the manuscript. 

This research is supported in part by JSPS KAKENHI Grant Number JP.21H03425. 

However, funding information should not appear in the Acknowledgments section or other areas of your manuscript. We will only publish funding information present in the Funding Statement section of the online submission form. 

YH is partially supported by a Grant-in-Aid for Scientific Research (Grant Number JP.21H03425) from the Japan Society for the Promotion of Science. The funders had no role in study design, data collection and analysis, decision to publish, or preparation of the manuscript. 

Reviewers' comments:

Reviewer's Responses to Questions

**Comments to the Author**

1. Is the manuscript technically sound, and do the data support the conclusions?

Reviewer #1: Yes

Reviewer #2: No

2. Has the statistical analysis been performed appropriately and rigorously? 

Reviewer #1: Yes

Reviewer #2: No

3. Have the authors made all data underlying the findings in their manuscript fully available?

Reviewer #1: Yes

Reviewer #2: Yes

4. Is the manuscript presented in an intelligible fashion and written in standard English?

Reviewer #1: Yes

Reviewer #2: Yes

5. Review Comments to the Author

Reviewer #1: The authors start from the premise that manu real-world networks characterized by power-law degree distributions and claim that those are extremely vulnerable against malicious attacks. To obtain higher robustness  some link addition methods were proposed. Precisely, two effective strategies for selecting nodes to add links are: the minimum degree and longest distance strategies.

So, the authors investigate the contributions of these strategies to improving the robustness by adding links in distinguishing the effects of degrees and distances as much as possible. Through

numerical simulation, they find that the robustness is effectively improved by adding links on the minimum degree strategy for both synthetic trees and real networks. Here are some suggestions for improvement:

(1) Is there any reason that the authors only consider trees in the synthetic study? Even though they are fragile against attacks, non-tree graph structures should also be analyzed since real-world networks are usually not trees.(2) If more links are continuously added to the attacked graph (after exceeding 2000 and 3000 as the limit of x-axis in the figures), can we observe convergence of the robustness?(3) In the synthetic experiment section, the authors only consider graphs with 1000 nodes. I suggest also adding an experiment which investigates how the graph size and density influences the robustness and efficiency in the same setting. If such behaviors are expected to be consistent across graphs which have different sizes and densities, please further explain it.(4) Real world networks display complex topologies and there has been significant work recently to characterize the network properties from many angles such as dealing with the power laws and fractal features and exploit them for network reconstruct, or for inferring network generating rules, or for quantifying the degree of network complexity. Here are some such network models "Reconstructing missing complex networks against adversarial interventions." Nature communications 10, no. 1 (2019): 1-12. "Network science characteristics of brain-derived neuronal cultures deciphered from quantitative phase imaging data." Scientific reports 10, no. 1 (2020): 1-13. "Hidden network generating rules from partially observed complex networks." Communications Physics 4, no. 1 (2021): 1-12. "Deciphering the generating rules and functionalities of complex networks." Scientific reports 11, no. 1 (2021): 1-15. that match real world networks and should be discussed at least if not considered as case studies as more realistic network models.

(5) It's good that the authors mention trees are more fragile against attacks. I wonder how this is reflected in the experiments?This is another reason why the authors can also consider non-tree and more complex simulated graphs. It would be great if the authors can measure and quantify the similarity/dissimilarity of the arbitrary simulated graphs with tree-like graphs, and further show how the tree structure influences the link addition methods and the corresponding robustness.

Reviewer #2: The current work investigates the improvement of network robustness and efficiency via adding links based on minimum degree and longest distance. Concretely, it proposes serval strategies of adding links for selecting nodes with minimum-k degree and longest distance, and improves the network density, which makes the varying network be more robustness and efficiency. In my view, the strategy of adding links is common method for improving network robustness and efficiency, and has be developed for a long time. In the current work, the related works are out-of-date (most of them before 2019, except of the authors’ works [11] and [15]), as well as the comparison methods. Thus, the authors should survey more recent works, and introduce them to compare with the proposed method. Additionally, it does not illustrate the difference of the proposed method from the authors’ works [11] and [15]. Most importantly, my comments involve the figures and the corresponding results, shown as follows as:

(1) Figures are blurred especially when they are enlarged.

(2) In Figures 2-7, and 9-10 the strategies of min-k RandAdd and min-k LongAdd have the same ability of improving network robustness and efficiency. These results suggest that the min-k selecting strategy play the key operation, thus I think that the authors should check the results and independently discus the min-k selecting strategy.

6. PLOS authors have the option to publish the peer review history of their article (what does this mean?). If published, this will include your full peer review and any attached files.

Reviewer #1: No

Reviewer #2: No

---

## [Author Response · Author response to Decision Letter 0]

27 Sep 2022

Prof. Lazaros K. Gallos

Academic Editor

PLOS ONE

Dear Prof. Lazaros K. Gallos,

Re: Manuscript reference No. PONE-D-22-09998

Thank you for your ongoing consideration of our manuscript entitled “Adding links on minimum degree and longest distance strategies for improving network robustness and efficiency” for publication in PLOS ONE. 

We also appreciate the time and effort you and the reviewers have dedicated to providing insightful feedback on ways to improve our paper. We have incorporated changes that reflect the detailed suggestions. We hope that our revised manuscript and the responses satisfactorily address all the issues and concerns you and the reviewers have pointed out.

In the revised manuscript with track changes, we marked added text in blue and removed ones in red. In accordance with suggestions, we added several results on non-tree scale-free networks as more realistic networks (described in 241-308 lines of the revised manuscript with track changes). We inserted new Figures 4-9, and the numbers of figures are updated. 

To facilitate your review of our revisions, the following pages are our point-by-point responses to the questions and comments.

Again, thank you for inviting us the opportunity to improve our manuscript with your valuable comments and queries. We have worked hard to incorporate your feedback and hope that these revisions persuade you to accept our submission.

Sincerely,

Masaki Chujyo

Ph. D. student 

Japan Advanced Institute of Science and Technology

1-1 Asahidai, Nomi, Ishikawa 923-1292 Japan

E-mail: mchujyo@jaist.ac.jp

On behalf of all authors.

For the comments of Reviewer 1

(1) Is there any reason that the authors only consider trees in the synthetic study? Even though they are fragile against attacks, non-tree graph structures should also be analyzed since real-world networks are usually not trees.

Our answer (1)

 Thank you for the suggestion. Yes, there is no reason for the restriction on trees in the synthetic study. Moreover, we added some results of non-tree scale-free networks in 241-308 lines (blue texts) of the revised manuscript with track changes. We discussed that the min-k ShortAdd method improves the robustness more effectively for non-tree networks. 

(2) If more links are continuously added to the attacked graph (after exceeding 2000 and 3000 as the limit of x-axis in the figures), can we observe convergence of the robustness?

Our answer (2)

 Yes. We observe the convergence of the robustness for scale-free networks (Fig 5), in comparing the ratios of added and existing links. As the number of added links increases, the slope of improved robustness becomes smaller. For a sufficiently large number of added links, the differences between the methods become small through improving the robustness. We described it in 244-254 lines (blue texts) of the revised manuscript with track changes. 

(3) In the synthetic experiment section, the authors only consider graphs with 1000 nodes. I suggest also adding an experiment which investigates how the graph size and density influences the robustness and efficiency in the same setting. If such behaviors are expected to be consistent across graphs which have different sizes and densities, please further explain it.

Our answer (3) 

 We added the results of the initial scale-free networks with 500, 1000, and 5000 nodes (Fig 5). When we compared the robustness for the ratios of added and existing links, there was little difference according to node size. We described it in 244-254 lines (blue texts) of the revised manuscript with track changes.

 As density, we considered both the initial average degree and the ratio of added links. For the initial average degree, we added the results of the initial scale-free networks with average degrees 2 and 4 (Fig 5). We found that the min-k ShortAdd method is effective for non-tree scale-free networks (the initial average degree 4), although it is inefficient for trees (the initial average degree 2). We described it in 254-268 lines (blue texts) of the revised manuscript with track changes.

 For the ratio of added links, in general, the robustness increases, as the number of added links increases. In comparing the slopes of increasing robustness, the difference of slopes for each method becomes smaller, as the number of added links increases (Fig 5A). For sufficiently large numbers of added links, the ratios of added links are more dominant for the improvement of robustness more the differences between methods. We described it in 248-250 lines (blue texts) of the revised manuscript with track changes.

(4) Real world networks display complex topologies and there has been significant work recently to characterize the network properties from many angles such as dealing with the power laws and fractal features and exploit them for network reconstruct, or for inferring network generating rules, or for quantifying the degree of network complexity. Here are some such network models "Reconstructing missing complex networks against adversarial interventions." Nature communications 10, no. 1 (2019): 1-12. "Network science characteristics of brain-derived neuronal cultures deciphered from quantitative phase imaging data." Scientific reports 10, no. 1 (2020): 1-13. "Hidden network generating rules from partially observed complex networks." Communications Physics 4, no. 1 (2021): 1-12. "Deciphering the generating rules and functionalities of complex networks." Scientific reports 11, no. 1 (2021): 1-15. that match real world networks and should be discussed at least if not considered as case studies as more realistic network models.

Our answer (4) 

 According to your suggestion, we described that the extension of our methods to more complicated and realistic networks will be important in the future in 418-420 lines (blue texts) of the revised manuscript with track changes.

(5) It's good that the authors mention trees are more fragile against attacks. I wonder how this is reflected in the experiments? This is another reason why the authors can also consider non-tree and more complex simulated graphs. It would be great if the authors can measure and quantify the similarity/dissimilarity of the arbitrary simulated graphs with tree-like graphs, and further show how the tree structure influences the link addition methods and the corresponding robustness.

Our answer (5). 

 We do not directly show that the trees are vulnerable to attacks in our experiments, but we expected to show the difference of improving robustness by each method with a small number of added links. For trees, we conclude that the min-k RandAdd and min-k LongAdd methods are effective for a larger number of added links, while the LongAdd method is effective for a smaller number of added links.

 Although we do not discuss the similarity to trees, we show the difference in improvement in robustness depending on whether the initial network is a tree or not, by comparing the results of trees or non-trees. We described it in 254-268 lines (blue texts) of the revised manuscript with track changes.

For the comments of Reviewer 2

(1) Figures are blurred especially when they are enlarged.

Our answer (1)

 We changed the format of the figures from tiff to eps in order to be more clear.

(2) In Figures 2-7, and 9-10 the strategies of min-k RandAdd and min-k LongAdd have the same ability of improving network robustness and efficiency. These results suggest that the min-k selecting strategy play the key operation, thus I think that the authors should check the results and independently discus the min-k selecting strategy.

Our answer (2)

 Since the min-k RandAdd and min-k LongAdd methods select nodes with the minimum degree, both degree distributions and the variances are the same, as shown in Figs 6 and 7. These methods result in homogenous degree distributions, which are dominant in the improvement of robustness. However, the distances of selected node pairs are different. We show that the distances are larger for the min-k LongAdd method in Fig 8. According to your valuable suggestion, we added the discussion on the minimum degree strategy in 269-308 lines (blue texts) of the revised manuscript with track changes.

---

## [Decision Letter · Decision Letter 1]

13 Oct 2022

Adding links on minimum degree and longest distance strategies for improving network robustness and efficiency

PONE-D-22-09998R1

Dear Dr. Chujyo,

We’re pleased to inform you that your manuscript has been judged scientifically suitable for publication and will be formally accepted for publication once it meets all outstanding technical requirements.

Kind regards,

Lazaros K. Gallos

Academic Editor

PLOS ONE

Additional Editor Comments (optional):

Reviewers' comments:

Reviewer's Responses to Questions

**Comments to the Author**

1. If the authors have adequately addressed your comments raised in a previous round of review and you feel that this manuscript is now acceptable for publication, you may indicate that here to bypass the “Comments to the Author” section, enter your conflict of interest statement in the “Confidential to Editor” section, and submit your "Accept" recommendation.

Reviewer #1: All comments have been addressed

Reviewer #2: All comments have been addressed

2. Is the manuscript technically sound, and do the data support the conclusions?

Reviewer #1: Yes

Reviewer #2: Yes

3. Has the statistical analysis been performed appropriately and rigorously? 

Reviewer #1: Yes

Reviewer #2: Yes

4. Have the authors made all data underlying the findings in their manuscript fully available?

Reviewer #1: Yes

Reviewer #2: Yes

5. Is the manuscript presented in an intelligible fashion and written in standard English?

Reviewer #1: Yes

Reviewer #2: Yes

6. Review Comments to the Author

Reviewer #1: Thank you very much for addressing all my comments and thoughts. I think this is an interesting study that should receive more attention from the network science community.

Reviewer #2: (No Response)

7. PLOS authors have the option to publish the peer review history of their article (what does this mean?). If published, this will include your full peer review and any attached files.

Reviewer #1: No

Reviewer #2: No

---

## [Editor Report · Acceptance letter]

17 Oct 2022

PONE-D-22-09998R1 

Adding links on minimum degree and longest distance strategies for improving network robustness and efficiency 

Dear Dr. Chujyo:

I'm pleased to inform you that your manuscript has been deemed suitable for publication in PLOS ONE. Congratulations! Your manuscript is now with our production department. 

Kind regards, 

on behalf of

Dr. Lazaros K. Gallos 

Academic Editor

PLOS ONE